# Comparison of High- and Low-LET Radiation-Induced DNA Double-Strand Break Processing in Living Cells

**DOI:** 10.3390/ijms21186602

**Published:** 2020-09-09

**Authors:** Stefan J. Roobol, Irene van den Bent, Wiggert A. van Cappellen, Tsion E. Abraham, Maarten W. Paul, Roland Kanaar, Adriaan B. Houtsmuller, Dik C. van Gent, Jeroen Essers

**Affiliations:** 1Department of Molecular Genetics, Erasmus University Medical Center, 3015 GD Rotterdam, The Netherlands; s.roobol@erasmusmc.nl (S.J.R.); i.vandenbent@student.tudelft.nl (I.v.d.B.); m.w.paul@erasmusmc.nl (M.W.P.); r.kanaar@erasmusmc.nl (R.K.); d.vangent@erasmusmc.nl (D.C.v.G.); 2Oncode Institute, Erasmus University Medical Center, 3015 GD Rotterdam, The Netherlands; 3Department of Radiology & Nuclear Medicine, Erasmus University Medical Center, 3015 GD Rotterdam, The Netherlands; 4Optical Imaging Center (OIC), Erasmus University Medical Center, 3015 GD Rotterdam, The Netherlands; w.vancappellen@erasmusmc.nl (W.A.v.C.); t.abraham@erasmusmc.nl (T.E.A.); a.houtsmuller@erasmusmc.nl (A.B.H.); 5Department of Vascular Surgery, Erasmus University Medical Center, 3015 GD Rotterdam, The Netherlands; 6Department of Radiation Oncology, Erasmus University Medical Center, 3015 GD Rotterdam, The Netherlands

**Keywords:** DNA double-strand breaks, high linear energy transfer, alpha particles, homologous recombination, live-cell microscopy, nonhomologous DNA end-joining

## Abstract

High-linear-energy-transfer (LET) radiation is more lethal than similar doses of low-LET radiation types, probably a result of the condensed energy deposition pattern of high-LET radiation. Here, we compare high-LET α-particle to low-LET X-ray irradiation and monitor double-strand break (DSB) processing. Live-cell microscopy was used to monitor DNA double-strand breaks (DSBs), marked by p53-binding protein 1 (53BP1). In addition, the accumulation of the endogenous 53BP1 and replication protein A (RPA) DSB processing proteins was analyzed by immunofluorescence. In contrast to α-particle-induced 53BP1 foci, X-ray-induced foci were resolved quickly and more dynamically as they showed an increase in 53BP1 protein accumulation and size. In addition, the number of individual 53BP1 and RPA foci was higher after X-ray irradiation, while focus intensity was higher after α-particle irradiation. Interestingly, 53BP1 foci induced by α-particles contained multiple RPA foci, suggesting multiple individual resection events, which was not observed after X-ray irradiation. We conclude that high-LET α-particles cause closely interspaced DSBs leading to high local concentrations of repair proteins. Our results point toward a change in DNA damage processing toward DNA end-resection and homologous recombination, possibly due to the depletion of soluble protein in the nucleoplasm. The combination of closely interspaced DSBs and perturbed DNA damage processing could be an explanation for the increased relative biological effectiveness (RBE) of high-LET α-particles compared to X-ray irradiation.

## 1. Introduction

Double-strand breaks (DSBs) are considered the most dangerous type of DNA damage, as they lead to cell death or mutations if left unrepaired [1]. Recognition of DSBs is the first step toward repair. The first response of the cell is to initiate a highly complex DNA damage response (DDR), in which lesions are identified and marked, thereby initiating DNA repair pathways [2,3]. Activation of DNA repair pathways involves the modification of histones, leading to specific histone marks. This information eventually results in chromatin remodeling [4]. As a result, the histone modifications flanking the DSB are expanded, which leads to the recruitment of DNA damage response proteins like p53-binding protein 1 (53BP1), which can be visualized as nuclear foci [5]. The accumulation of 53BP1 requires the direct recognition of a DSB-specific histone code (H4-K20me2 [6]) and can, therefore, be used as a surrogate marker for DSBs. In addition, 53BP1 influences the DNA repair pathway choice by antagonizing long-range DNA end-resection [4].

There are two major pathways by which DSBs can be repaired, nonhomologous end-joining (NHEJ) and homologous recombination (HR) [7], each of which comprises a number of subpathways. The most direct way to repair DSBs is via the NHEJ pathway, which is active throughout the cell cycle. After DSBs are recognized, DNA ends are processed and eventually rejoined by DNA ligase IV [8]. In the S/G2 phases of the cell cycle, HR is active as an additional repair mechanism. In these phases of the cell cycle, the sister chromatid serves as a template for repair [9]. During HR, the DSB ends are resected by 5′–3′ exonucleases, and single-stranded DNA is stabilized by RPA [10]. Replacement with recombination protein RAD51 facilitates the search for the homologous sister chromatid and error-free repair [11,12].

Different DSB repair pathways have evolved because not all breaks are equal, with the difference between the repair of one-ended and two-ended breaks being the most obvious; however, the chemical nature of DNA ends also demands different end-processing factors [7]. By altering radiation types, this difference can be assessed [13]. Differences in ionizing radiation (IR)-induced DSBs are mostly the result of radiation with high or low linear energy transfer (LET). The LET of IR describes the amount of energy which is deposited per unit of length in the material it passes through, for example, tissue [14]. X-ray and gamma irradiation are characterized by a low-LET, inducing sparse and mostly single-strand DNA breaks (SSBs). In contrast, high-LET α-particles or heavy ions result in very localized DNA damage containing a large amount of DSBs [15,16,17,18]. The difference in LET has a direct effect on the DDR of the treated cells. High-LET IR mostly induces fewer but larger DSB foci per unit of dose, suggesting multiple DSBs in one focus [19]. In addition, the nuclear DSB focus resolution is slow after high-LET IR compared to low-LET IR, indicating that DSBs are processed in a different way and/or with different kinetics [20,21].

Most current knowledge was gathered in formaldehyde-fixed cells and immunostaining of DSB foci at various time-points. However, fixed samples provide limited options to obtain information regarding the spatial-temporal behavior of individual DSB foci. The implementation of live-cell imaging overcomes these limitations [19,22,23,24,25]. Dynamic live-cell imaging offers a more dynamic view of DSB processing, including the mobility of surrounding chromatin [26].

A previous study showed that α-particle-induced 53BP1 foci are more persistent and larger, and show high mobility compared to X-ray irradiation-induced 53BP1 foci [19]. To determine the kinetics and mechanism of DSB processing after high-LET α-particle irradiation compared to low-LET X-ray irradiation, we repeated the experiment, increased the timeframe, and investigated DSB repair protein behavior. We observed 53BP1 foci disappearing within a few hours after X-ray irradiation, and the remaining foci showed an increase in size and intensity. In contrast, α-particle-irradiated cells showed mainly persistent foci, and remaining foci did not show any changes in size or intensity. In addition, by combining immunostaining for 53BP1 and RPA (indicative for resection), we found that α-particle-induced foci contained multiple DSBs that tend to have a relatively high probability of resection.

## 2. Results

### 2.1. Focus Segmentation for Comparison of DSB Processing between High- and Low-LET Irradiation

To study the progression of DSB foci over time, live-cell microscopy provides an abundance of information compared to fixed time-point analysis, such as real-time mobility [25]. Using U2OS cells which stably expressed 53BP1-GFP, we analyzed 53BP1-GFP focus dynamics after irradiation with X-rays or α-particles. Irradiated cells were imaged using confocal microscopy. Images were taken 20 min apart over a 17 h time period post irradiation. These image sequences were subjected to an image processing pipeline to extract the real-time changes of 53BP1-GFP foci.

First, cell nuclei were segmented and subjected to a stabilizing correction to compensate for any changes in nucleus orientation or shape. This ensured analysis of focus dynamics independent of cellular movement (Appendix A). Subsequently, images were further processed by separating neighboring foci, using the watershed tool in ImageJ (see Section 4). This provided a dataset of single 53BP1-GFP foci, which could be followed over time (Figure 1A).

The segmentation of foci in each consecutive frame allowed tracking of individual 53BP1-GFP foci in time. With the use of an algorithm, on the basis of a threshold of 0.7-µm change, the individual foci were linked together between consecutive frames, which formed focus tracks in time. The linking process was based on the distance between the center of mass (COM) points of the focus in consecutive frames (Figure 1B,C). Focus tracks generated from α-particle-irradiated and X-ray-irradiated cells were analyzed. In order to evaluate the quality of our segmentation method, we analyzed the correlation between manual and script-based counting measurements of 10 random cell nuclei. The correlation plots showed good correlation for α-particle-irradiated cells (*R*^2^ = 0.89) and X-ray-irradiated cells (*R*^2^ = 0.82) (Figure 1D,E). Using this newly developed semi-automated analysis, we compared the DSB processing dynamics, marked by 53BP1-GFP, of α-particle-irradiated cells with X-ray-irradiated cells.

### 2.2. α-Particle-Induced 53BP1-GFP Foci Are Slowly Resolved

The formation and resolving of 53BP1 foci can be used as a surrogate marker to determine the kinetics of DSB repair [4]. To investigate the kinetics of DSB repair in live cells, we analyzed focus tracks to measure how individual foci appeared or disappeared and changed position, size, or intensity after irradiation. First, by measuring how many foci are present at certain time-points, we determined the number of foci per nucleus over time.

At the start of imaging, 15 min after irradiation, both treatments showed a similar number of 53BP1-GFP foci per nucleus. In addition, after both treatments, the number of 53BP1-GFP was significantly higher compared to nontreated cells in which an average of 2.3 ± 0.15 53BP1-GFP foci per nucleus were observed in the investigated timeframe. However, in α-particle-irradiated cells, this number doubled after 2 h to 14 foci per nucleus. In contrast, X-ray-irradiated cells showed a reduction in foci per nucleus from 2 h after irradiation onward down to three foci per nucleus. Moreover, the number of foci per nucleus in α-particle-irradiated cells did not decline until 13 h after irradiation and was significantly higher compared to X-ray-irradiated cells between 200 and 960 min after irradiation (Figure 2A). By following the individual foci over time, we could determine how long a focus was visible after appearance. This was quantified and referred to as the duration of the focus track (Figure 2B). The focus tracks were binned into groups with durations of 20–80, 100–160, 180–240, 260–320, and 340–400 min. We found that, for both treatments, most of the segmented foci were visible between 20 and 80 min. Interestingly, α-particle-irradiated cells had more focus tracks which were observed for a longer time than X-ray-irradiated cells.

### 2.3. The Mobility of 53BP1-GFP Foci Is Similar after High- and Low-LET Irradiation

Increased mobility of chromatin surrounding DNA damage has been reported and was suggested to affect DNA repair [27]. As we observed that DSB focus resolving was different in α-particle-irradiated cells compared to X-ray-irradiated cells, we were interested if chromatin mobility might have been affected as well. To compare chromatin mobility, we directly measured the mobility of 53BP1-GFP foci upon treatment of the cells with the different irradiation qualities. By using the same binned groups as before, we investigated differences between foci which were visible for a short or long period of time.

First, we determined the mean square displacement (MSD) which was plotted per track duration group (Figure 2C–F and Appendix A). The curvature of an MSD plot can indicate whether the mobility of foci is apparent diffusion (linear line) or confined motion (curved plot). We observed no difference in mobility of 53BP1-GFP foci when we compared α-particle-irradiated cells to X-ray-irradiated cells. The curvature of tracks which were shorter than 100 min showed a linear MSD plot, while tracks which were longer than 100 min showed a curved MSD plot. Moreover, the curvature of MSD plots was more apparent with increasing focus track length (Appendix A).

The slope of MSD curves defines the apparent diffusion coefficient or rate of movement. By fitting all the MSD curves on the first time intervals, we compared the diffusion coefficients between different track lengths. We observed no significant differences in the apparent diffusion coefficient of 53BP1-GFP foci after both treatments (Figure 2G). Interestingly, with increasing track length, the diffusion coefficient declined, showing that foci which were followed for a short time moved faster compared to foci which were followed for a longer time. These results suggest that there is no significant difference in mobility of 53BP1-GFP foci after X-ray or α-particle irradiation. However, short-lived foci seemed to show more diffusive behavior while long-lived foci showed confined motion.

### 2.4. 53BP1-GFP Protein Concentration and Focus Size Increase after X-ray Irradiation

Focus characteristics such as the intensity of 53BP1-GFP and focus size potentially reveal how DSB processing progresses over time [28]. In addition, focus size might give an indication of the extent of DNA damage. Pixel intensity can be used as a measure of the amount of fluorescent molecules, present at sites of DNA damage [29]. Therefore, we calculated the average pixel intensity within 53BP1-GFP foci.

The average pixel intensity of 53BP1-GFP foci showed similar values at the start of imaging, for both treatments. After 400 min, an increase in intensity of 53BP1-GFP foci was observed in X-ray-irradiated cells, but not in α-particle-irradiated cells (Figure 3A). The large difference in focus intensity and increased error bars could have been caused by a change in population distributions of the analyzed foci. Therefore, we generated distribution plots of the average pixel intensity at the start of imaging (0 min), and after 300 min, 600 min, and 900 min (Figure 3B–E). Indeed, we observed a gradual shift toward higher average pixel intensities of foci induced by X-rays (Appendix A). Interestingly, at 900 min after irradiation, two clear populations were present, which did not arise in the α-particle-irradiated cells (Figure 3E).

Average pixel intensity indicates differences in 53BP1-GFP molecules present per volume, but does not reflect the area in which the chromatin might be damaged. Therefore, we analyzed the area of individual foci over time. We observed that X-ray-induced 53BP1-GFP foci showed a sudden growth after 500 min, increasing from 0.8 to 1.3 µm^2^ (Figure 3F and Appendix A). Interestingly, the foci in α-particle-irradiated cells were initially larger compared to foci in X-ray-irradiated cells (1.1 vs. 0.8 µm^2^), but showed little or no increase in size over time.

The increase in focus size does not necessarily imply that the intensity of 53BP1-GFP increases at a similar rate. For example, when a focus increases in size but no additional 53BP1 is recruited, this focus decreases in average pixel intensity. Indeed, we observed a linear increase in average pixel intensity per focus from the start of imaging after X-ray irradiation, but focus growth only occurred after 500 min. Multiplying the average pixel intensity by the focus area leads to the total amount of 53BP1-GFP present in the focus. Using this parameter, the focus growth can be related to the corresponding intensity, indicating whether more protein is recruited to the visible focus or not.

The initial amount of 53BP1-GFP per focus was similar for both α-particle- and X-ray-irradiated cells. However, the amount of 53BP1-GFP in X-ray-induced foci showed an increase over time (Figure 3G and Appendix A). For α-particle-induced foci, the amount of 53BP1-GFP showed no change until later than 900 min, where the amount slightly decreased, possibly due to fluorescent bleaching. We conclude that 53BP1-GFP foci increase more in intensity and size after X-ray irradiation than after α-particle irradiation.

### 2.5. Dose-Dependent Characteristics of Foci in EdU-Positive Cells after Both Treatments

The differences in foci intensity and size over time after X-ray or α-particles could be related to differences in DNA damage load. Therefore, we performed a dose gradient experiment using both α-particle and X-ray irradiation, ranging from 0.5 to 6 Gy and fixed cells 1 h post irradiation. In addition, to investigate functional protein and minimize the interference of overexpression constructs, cells were stained for endogenous 53BP1 and RPA as markers for DSBs and resection, respectively (Figure 4A,B). Quantification of both 53BP1 and RPA was done by creating a segmentation mask using standard thresholds (see Section 4). The segmentation provided the number of foci, and the signal within the mask could be used to determine the average pixel intensity of a focus.

The number of 53BP1 foci per cell increased in a dose-dependent manner for both treatments (Figure 4C). X-ray irradiation resulted in more 53BP1 foci per cell compared to α-particle-irradiated cells for all doses. Interestingly, 53BP1 focus intensity decreased in a dose-dependent manner after α-particle irradiation compared to little to no change after X-ray irradiation (Figure 4D). Moreover, the intensity of 53BP1 foci was fourfold higher after induction by α-particles than X-rays at a dose of 0.5 Gy.

Subsequently, we investigated the reason for the differences in protein recruitment after α-particle and X-ray irradiation. We considered that different DSB repair pathways might be activated. In addition, the extent of resection at the DNA ends determines what pathway is activated for DSB repair. Therefore, we quantified RPA foci in EdU-positive cells to investigate possible differences in resection events.

In X-ray-irradiated cells the number of RPA foci per cell increased significantly at a dose of 4.5 Gy and higher (Figure 4A,E). In addition, the average RPA focus intensity showed a dose-dependent increase, only at a dose of 4.5 or higher (Figure 4F). This suggests that, at this dose and above, DSBs seemed to be more susceptible to resection after X-ray irradiation. Interestingly, after α-particle irradiation, the number of RPA foci per cell did not increase until the dose of 5.5 Gy (Figure 4E). However, the average RPA intensity of foci induced by α-particles showed an increase at 4 Gy, the dose at which the number of foci increased in X-ray irradiated cells (Figure 4F).

The high accumulation of 53BP1 and RPA protein in a single focus after α-particle irradiation suggests that there is more extensive resection or that multiple DNA ends are present in one focus.

### 2.6. α-Particle-Induced 53BP1 Foci Show Multiple Individual Resection Events

Subsequently, we investigated whether the observed RPA foci colocalized with the 53BP1 foci. The location of RPA and 53BP1 foci showed differences after α-particle or X-ray irradiation (Figure 5A). RPA foci induced by X-rays did not colocalize with 53BP1 foci and appeared mutually exclusive, while, after α-particle irradiation, we observed mixed 53BP1/RPA foci. To quantify this observation, we used the segmentation mask of 53BP1 foci to measure the pixel intensity of RPA within these foci. The average pixel intensity of RPA colocalizing with 53BP1 foci was significantly higher after α-particle treatment than after X-ray irradiation (Figure 5B).

With the use of structured illumination microscopy (SIM), we obtained higher-resolution images, doubled in the *x-*, *y*-, and *z*-axis, to zoom in on individual 53BP1 foci. Using the same segmentation masks of 53BP1 as above, we identified individual RPA foci within a 53BP1 focus. We observed 40% of α-particle-induced 53BP1 foci with one or more colocalizing RPA foci, compared to 5% of the X-ray-induced foci (Figure 5C). Moreover, none of the 53BP1 foci in X-ray-irradiated cells had more than two RPA foci. Notably, the enhanced resolution of SIM revealed exclusion of 53BP1 at RPA foci at the nanoscale level (Figure 5D). These results suggest that multiple individual DNA ends are present in one focus after α-particle irradiation, of which one or more can be resected followed by RPA protein accumulation.

## 3. Discussion

Cellular survival is impacted more after receiving high-LET radiation types compared to low-LET radiation. The increased biological effectiveness of high-LET radiation is thought to be the result of the highly condensed energy deposition pattern. To study how cells cope with high- or low-LET radiation types, we compared α-particle irradiation (high LET) to X-ray irradiation (low LET) using live-cell imaging. We observed differential processing of DSBs after α-particle irradiation compared to X-ray irradiation. Cells treated with α-particles show slow DSB repair and differences in 53BP1-GFP accumulation and focus growth. Endogenous focal accumulation of the proteins 53BP1 and RPA was much higher after α-particles than X-rays. The high accumulation of these markers in α-particle-irradiated cells suggests multiple individual resection events within single DSB foci.

Our observations highlight a substantial difference in DSB processing after X-ray irradiation compared to α-particle irradiation. Employing live-cell imaging to compare α-particle- to X-ray-irradiated cells revealed large differences in the processing of 53BP1-GFP foci. We found that foci induced by α-particles (a) are eliminated more slowly compared to X-ray-induced foci, (b) show no average pixel intensity increase over time, in contrast with X-ray-induced foci, (c) are initially larger than X-ray-induced foci but show no increase in size, which is prominent in X-ray induced foci, and (d) accumulate larger amounts of 53BP1-GFP in foci than X-ray-induced foci.

The repair of induced DSBs in α-particle-irradiated cells appears to be minimal and slow, while X-ray-induced DSBs are repaired quickly, most likely via NHEJ [30]. However, repair via NHEJ can be divided into a fast (in euchromatin) and slow (in heterochromatin) component [31,32]. The slow component of NHEJ repair involves local chromatin decondensation, regulated by factors such as 53BP1, ATM, and KAP-1 [18,31,33].

Chromatin decondensation could be the factor which causes a delay in DSB processing, which we observed after X-ray irradiation (Figure 3F). Additionally, chromatin decondensation could widen the area around DSBs, a possible explanation for 53BP1 focus size increase. Interestingly, during DSB focus enlargement, we observed an increase in 53BP1-GFP intensity per focus. This is contradictory to our argument of chromatin widening, which would be expected to result in a dilution of GFP signal. However, this is not necessarily true; fast resolution of 53BP1 foci in the early phase of DSB repair would result in increasing free 53BP1 in the nucleus, which could then relocate to the remaining slowly repaired DSBs, thereby increasing focus intensity. Indeed, we observed an increase in the total amount of 53BP1-GFP per focus in X-ray-irradiated cells, suggesting relocalization of 53BP1-GFP (Figure 3G).

Chromatin condensation and decondensation in response to DNA damage leads to large-scale reorganization of chromatin fibers [26]. The consequence of chromatin (de)condensation is the accessibility of many factors. Additionally, the loosening of chromatin could alter chromatin mobility. Numerous reports showed that DNA damage increases the mobility of chromatin and the induced DSBs, as elaborately reviewed in [26] and [34]. The highly condensed energy pattern of high-LET irradiation reduces efficient DNA repair and causes the chromosomal reorganization to be disrupted [18,35]. Another report showed higher DSB mobility after α-particle irradiation when compared with X-ray irradiation, using similar live-cell techniques to those in our study [19]. We observed no apparent difference in chromatin mobility between α-particle or X-ray irradiation. A possible explanation could be the difference between the 20-min interval (this report) and the 1-min interval [19]. Chromatin motion at different time scales can be different [36]. Therefore, comparing 20-min intervals to 1-min intervals would not be correct. To compare DSB mobility in our setting to other mobility studies within the chromatin, we should consider shorter time intervals. The difficulty lies in keeping the possibility to image for several hours; 1-min intervals for several hours could induce photo bleaching. A possible way to overcome this difficulty is to capture multiple frames with a short interval, every so often. This would capture the mobility of DSB foci whilst keeping the possibility to image for several hours.

We observed that most 53BP1 foci disappeared in time after X-ray irradiation, but not after α-particle irradiation, suggesting that high-LET DSBs are not repaired in the timeframe (17 h) of this experiment. The remaining question relates, therefore, to the cause of this delay of focus resolving compared to X-ray-induced damage.

Clustered or complex DNA damage is suggested to be an important cause of the increased biological effectiveness of high-LET radiation, and both terms are used to describe multiple types of DNA damage in close vicinity [17,37,38,39]. We observed multiple independent resection events, marked by RPA, localizing to single 53BP1 foci. These structures are best described as “closely interspaced DSBs”, leading to increased biological effectiveness when not properly repaired [23,40]. Indeed, super-resolution techniques previously uncovered “nanoclusters” in previously thought single DSB foci [37]. Closely interspaced DSBs would explain why we observed a relatively low number of 53BP1 and RPA foci, which have in turn high intensity. In addition, we could speculate that closely interspaced DSBs cause hyper-resection, causing a lack of repair in the G1 phase with no HR factors available, leading to persistent DSBs. Another possibility in G1 phase would be slow resection-dependent NHEJ, leading to microhomology-mediated end-joining [41].

53BP1 is active in a delicate balancing act to regulate the extent of resection in the S and G2 phases of the cell cycle. 53BP1 pool depletion by high-dose X-ray irradiation would limit the DNA end protection and lead to increased DNA resection [42,43]. Indeed, we found that the number and intensity of RPA foci increased with increasing X-ray dose, indicating that more resected DNA is present. In addition, 53BP1 intensity decreased with increasing α-particle dose, mimicking the 53BP1 depletion at high X-ray dose. The 53BP1 depletion is suggested to be caused by an abundance of DSBs, causing insufficient 53BP1 binding. Interestingly, the retention of 53BP1 is efficient up to 20–40 DSBs simultaneously, and exceeding this number may result in failing NHEJ and increased HR-directed repair [44]. Low α-particle dose could reach these numbers due to closely interspaced DSBs as a result of the condensed energy deposition pattern of high-LET radiation. Indeed, 53BP1 foci intensity decreased while RPA intensity increased after α-particle irradiation, suggesting impaired 53BP1 retention leading to resection (Figure 5). Interestingly, we observed a substantial increase in the number of RPA foci at a similar dose for X-ray and α-particle irradiation, suggesting that susceptibility to resection is not radiation type-dependent. However, the high RPA focus intensity after α-particles and colocalization with 53BP1, which is not observed after X-rays, might be an explanation for the difference in biological effectiveness between X-rays and α-particles.

The discussed results could be summarized in a working model in combination with the recent reports of 53BP1 exhaustion (Figure 6). We argue that the available 53BP1 pool is sufficient after 2 Gy of X-ray irradiation, leading to full coverage of the DNA ends and efficient DSB repair via NHEJ. At the later time-points, most DSBs are repaired, leading to increased numbers of 53BP1 molecules available for growth of the few remaining foci. However, the 53BP1 pool is insufficient at similar α-particle dose due to closely interspaced DSBs, leading to insufficient amounts of 53BP1 to cover all DNA ends, which allows resection. 53BP1 depletion becomes prominent and most DSBs are resected only at higher doses. In the S/G2 phase, this leads to HR-directed repair, whereas, in the G1 phase, repair would result in persistent DSBs.

In conclusion, our results indicate that the condensed energy deposition pattern of high-LET α-particles induces closely interspaced DSBs. The abundance of multiple DSBs in close vicinity throughout the cell nucleus leads to 53BP1 protein insufficiency and ineffective DNA end protection. This cascade of events might be an explanation of the increased biological effectiveness of high-LET α-particle irradiation, compared to low-LET X-ray irradiation.

## 4. Materials and Methods

### 4.1. Cell Culture

U2OS cells were cultured in Dulbecco’s modified Eagle medium (DMEM) supplemented with 1% penicillin/streptomycin (PS) and 10% fetal calf serum (FCS). Cells were incubated at 37 °C in a water-saturated atmosphere with 5% CO_2_. The 53BP1-GFP U2OS cell line was previously described and characterized [45]. In short, full-length m53BP1 was cloned into a pEGFP-C1 vector and transfected into U2OS cells.

### 4.2. X-ray and α-Particle Irradiation

Irradiated cells were cultured on round glass coverslips (diameter: 18 mm). X-ray irradiation was performed using the RS320 (Xstrahl Live Sciences), a self-contained cabinet, with a dose rate of 1.6554 Gy/min and working voltage of 195 kV and 10 mA. Alpha irradiation was performed as described before [46]. In brief, coverslips containing cultured cells were washed with PBS and placed upside down (cells facing down) on a Mylar dish. The Mylar dish was placed in the irradiation set-up with the cells facing down to the ^241^Am source. The center area of the coverslip was subsequently irradiated using alpha particles passing vertically through the cells, with the requested dose. In this study, the a-particles had an LET of 115 ± 10 keV/µm [47].

### 4.3. Immunofluorescence

Cells were washed with ice-cold PBS 1 h post irradiation. For RPA staining, cells were extracted with cold CSK buffer (10 mM HEPES-KOH, pH 7.9, 100 mM NaCl, 300 mM sucrose, 3 mM MgCl_2_, 1 mM EGTA, 0.5% (*v*/*v*) Triton X-100) and cold CSS buffer (10 mM Tris, pH 7.4, 10 mM NaCl, 3 mM MgCl_2_, 1% (*v*/*v*) Tween-20, 0.5% (*w*/*v*) sodium deoxycholate) for 5 min each before fixation in 4% PFA in PBS for 30 min at room temperature. Fixed cells were washed two times in PBS plus 0.1% Triton X-100 and washed 30 min in blocking solution (0.5% BSA plus 0.15% glycine in PBS). Primary rabbit anti-53BP1 (1:1000, Novus Biologicals, Centennial, CO, USA), mouse anti-RPA (1:1000, Calbiochem, San Diego, CA, USA), and mouse anti-yH2AX (1:1000, Millipore, Burlington, MA, USA) antibodies were diluted in blocking solution, and cells were incubated at 4 °C overnight. Hereafter, cells were washed two times for 10 min with PBS plus 0.1% Triton X-100 and washed shortly in blocking solution. Secondary antibodies Alexa Fluor goat anti-rabbit 594 and goat anti-mouse 488 (1:1000, ThermoFisher Scientific, Waltham, MA, USA) were diluted in blocking solution, and cells were incubated for 1 h at room temperature. S-phase cells were detected by EdU incorporation (10 µM, Merck, Darmstadt, Germany). Cells were labeled with EdU 30 min before fixation, which was detected using a Click-IT reaction according to the manufacturer’s protocol (Invitrogen, Waltham, MA, USA).

### 4.4. Confocal (Live-Cell) Imaging

To capture the progression of 53BP1-GFP foci, live-cell confocal microscopy was performed using a Leica SP5 confocal microscope. Immediately after irradiation (2 Gy for both X-ray and α-particles), coverslips were placed in a live-cell chamber and filled with prewarmed medium. The live-cell chamber was placed in a PeCon small chamber incubator with 37 °C, 5% CO_2_ regulation. Images of randomly selected groups of cells were acquired using a 40× HCX PL APO CS (NA = 1.25) oil objective at an interval of 20 min. GFP signal was detected using a laser line of 488 nm and emission filter of 500–550 nm. Three independent experiments were conducted capturing three distinct groups of cells at every experiment, for both treatments and for nontreated conditions.

To image the stained samples, a Leica SP5 confocal microscope was used. For each experiment, five images were acquired using a 40× HCX PL APO CS objective (NA = 1.25) and the appropriate laser lines and emission filters (DAPI/Atto Azide 390; excitation 405 nm, emission 435–480 nm, Alexa 488; excitation 488 nm, emission 500–550 nm, Alexa 594; excitation 561 nm, emission 570–630 nm). For image analysis, *z*-projections were made. Immunostained 53BP1 and RPA foci were quantified using ImageJ scripts (https://imagej.nih.gov/ij/download.html). In short, EdU-positive cells were segmented using auto-thresholds. Within the segmented nuclei, segmentation masks were made for individual foci using auto-thresholds and the watershed tool [48]. The mean intensity, area, and number of the segmented foci were measured using ImageJ.

### 4.5. Super Resolution Microscopy

Structured illumination microscopy (SIM) imaging was performed on a Zeiss Elyra PS1 with an Andor iXon DU 885 EMCCD camera (Carl Zeiss AG, Oberkochen, Germany) and 63× Plan Apochromat DIC oil lens (NA = 1.4) using 488- and 561-nm diode lasers with 100-ms exposure times. Samples were illuminated with a spatial line pattern that was shifted in five phases and rotated in five orientations. The raw images were reconstructed into a high-resolution three-dimensional (3D) dataset using the Zeiss 2012 PS1 ZEN software. RPA foci in SIM images were quantified manually.

### 4.6. Image Processing

Image sequences from live-cell experiments were analyzed in ImageJ using maximum-intensity projections. Using the 53BP1-GFP nuclear signal, cell nuclei were segmented for individual analysis. To minimize the translational and rotational motion of nuclei, a stabilizing correction was used on the basis of rigid body transformations (StackReg plugin by Philippe Thévenaz, Biomedical Imaging Group, Swiss Federal Institute of Technology Lausanne, https://imagej.net/StackReg). Single-image sequences containing single cells were subsequently analyzed to isolate and measure the 53BP1-GFP foci within nuclei. The first timeframe (*t* = 0) image containing the maximum projection of the *z*-stacks was duplicated and Gaussian-blurred with σ = 1. To segment foci within the nucleus, a lower threshold was manually set on this smoothed image of the first timeframe (with the upper threshold always being 255). Using the threshold option in ImageJ, an appropriate threshold was chosen, visually comparing an adjustable threshold image to the original input image. A visually accepted threshold led to an automatically calculated factor, depicted in Equation (1).
(1)$$factor=\frac{Manually\;set\;lower\;threshold-Cell\;mean\;intensity\;at\;t=0}{Cell\;standard\;deviation\;of\;mean\;intensity\;at\;t=0}$$
The threshold used on each consecutive timeframe (*t* ≥ 1) subsequently differed according to Equation (2).
(2)$$Threshold\left(t\right)=Cell\;mean\;intensity\;\left(t\right)+factor\times Cell\;standard\;deviation\;of\;mean\;intensity\;\left(t\right)$$

The factor was held constant for each frame of the same cell but could differ between cells to reach optimal 53BP1-GFP focus segmentation. If 53BP1-GFP foci were adherent due the threshold, the Watershed tool was used for foci separation [48]. The mean intensity, area, and coordinates of the center of mass of the segmented foci were measured using ImageJ.

### 4.7. Image Analysis

An algorithm was developed in MatLab to link individual foci between consecutive timeframes to form a focus track over time. The center of mass from one focus was compared to all the centers of mass from the foci in the consecutive timeframe. Based on the distance between the center of mass of a focus in one timeframe (*t* = *tn*) and a center of mass of a focus in a subsequent timeframe (*t* = *tn* + 1), foci were linked through time if the Euclidian distance between the center of masses was below 0.7 µm in two consecutive timeframes.

The track length was calculated based on the time interval used in the live-cell imaging and the number of frames the focus was visible in the track (Equation (3)).
(3)$$Track\;length\;\left(min\right)=\left(number\;of\;foci\;linked\;in\;track-1\right)\times time\;interval\left(=20\;min\right)$$

This process continued for the total number of timeframes imaged (in this case, 51 frames), equaling 1000 min (*t* = 0 was also a frame). The Matlab code was capable of detecting the splitting or merging of foci if one focus of a timeframe was linked to two distinct foci in the consecutive frame or if two distinct foci from one timeframe linked to the same focus in the consecutive frame, respectively. Through this process, a total of 2541 foci tracks were formed in 37 cells irradiated by α-particles and 970 foci tracks were formed in 26 cells irradiated by X-rays. Tracks were not allowed to have a gap, i.e., a focus had to be present in every consecutive frame. Additionally, tracks could start or end at any given time point in the image sequence.

Relative frequency plots were fitted by Kernel density estimates using the “fitdist” command in Matlab. The “kernel” option was applied, which fits a kernel to the data of a histogram. A smoothing bandwidth of 6, 0.3, and 220 was used for the kernel density estimations of the 53BP1 signal intensity per focus, focus area, and total 53BP1 on active foci, respectively. The area under the kernel density curve was normalized to be equal to the area of the histogram. The area of the histograms of α-particle- and X-ray-induced foci was scaled to be equal; hence, the areas under the curves were equal. A normalization factor was determined at *t* = 0 which made the highest peak of the histogram equal to 1, and this normalization factor was applied to each distribution at later time points.

The mean square displacement was calculated via the MSDanalyzer MatLab plugin [49]. Curves were fitted with the “fit” command in MatLab from the curve-fitting toolbox. The typical MSD curve for confined motion was linearly fit on a selected portion of the curve to yield an estimate for the diffusion coefficient D. As confined motion shows only after some time has passed, the first few data points of the MSD curve indicate diffusive motion, i.e., the curve is a straight line. The fit was made on the first four data points for each category.

The mean square displacement for diffusive motion in two dimensions (2D) is given by the Equation (4).
(4)$$MSD\left(t\right)=<r^{2}>=2dDt$$
where *d* = 2 is the dimension, and *D* is the diffusion coefficient.

This formula was approximated by the “fit type” command in MatLab, using independent *x* and dependent *y* with a fit method of “nonlinear least squares” and a start point in the origin. The approximation formula is shown in Equation (5), where a is calculated by Equation (6).
(5)$$y=f\left(x\right)=ax$$
(6)a=4D
This made it possible to calculate the diffusion coefficient directly from the fit as a value for the variable given by Matlab.

## Figures and Tables

**Figure 1 ijms-21-06602-f001:**
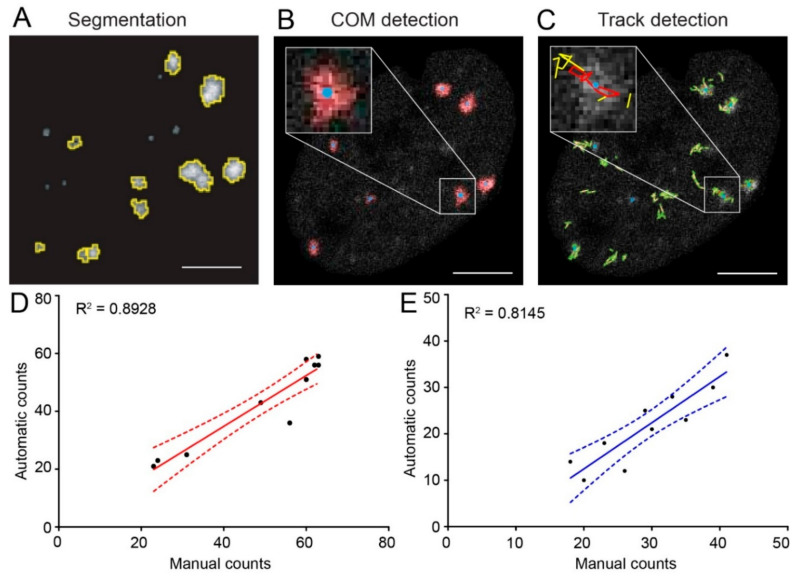
Overview of focus segmentation and focus track analysis. U2OS cells stably expressing p53-binding protein 1 (53BP1)-GFP were irradiated with 2 Gy of α-particle or X-ray irradiation. (**A**) ImageJ scripts were using for focus segmentation. (**B**) Center of mass (COM) per focus was calculated. (**C**) Center of mass was linked to consecutive frames (if COM was ≤ 0.7 µm). Tracks are indicated in red (active track at this time-point) or yellow (past track not active at this time-point). Scale bar indicates 5 µm. Segmentation of foci was correlated to manual counts of foci for both α-particle irradiation (**D**) and X-ray irradiation (**E**). For both treatments, three nonconsecutive frames were counted in 10 random nuclei. The averages of these counts are shown.

**Figure 2 ijms-21-06602-f002:**
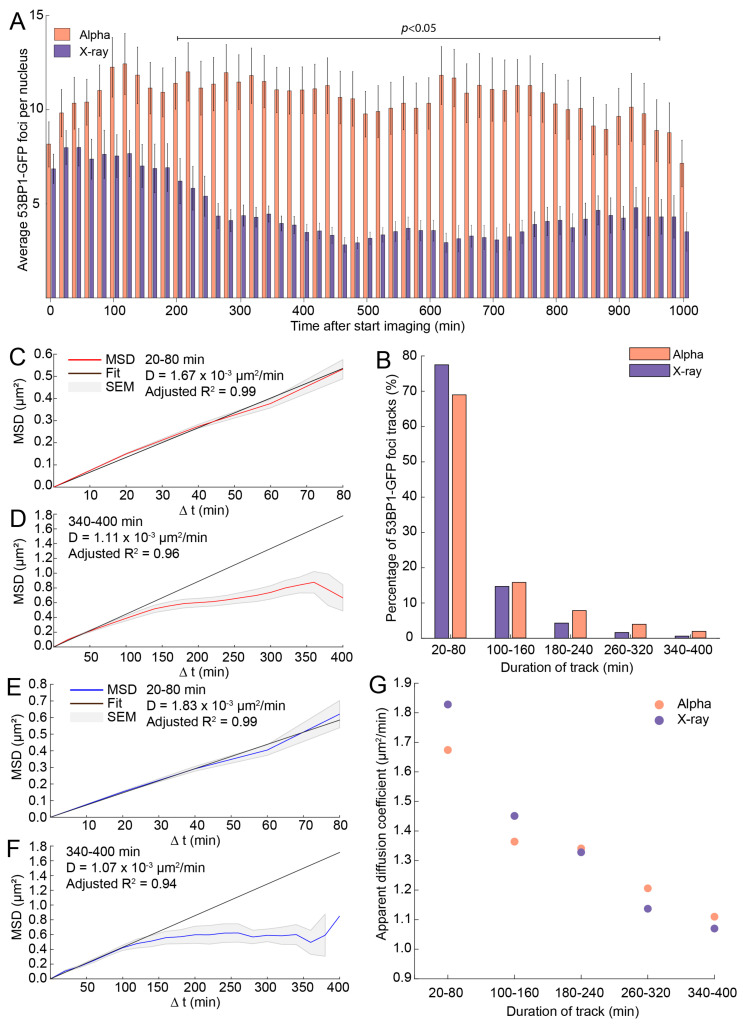
53BP1-GFP focus resolving after α-particles is slow compared to X-rays. (**A**) Average number of 53BP1-foci per cell over time. Foci in α-particle-irradiated (2 Gy) cells are depicted in red and those in X-ray-irradiated (2 Gy) cells are depicted in blue. All time-points between the indicated bar are significant differences between X-ray- and α-particle-irradiated cells (ANOVA, *p* < 0.05). Error bars indicate the standard error of the mean (SEM). (**B**) Percentage of detected 53BP1-GFP focus tracks that were present in α-particle-irradiated cells (red) or X-ray-irradiated cells (blue) for indicated time. (**C**–**F**) Mean square displacement (MSD) curves of the binned track lengths. α-particle-irradiated cells are depicted using a red line (**C**,**D**) and X-ray-irradiated cells are shown in blue (**E**–**G**). The average apparent diffusion coefficient of 53BP1-GFP foci, which was based on the fit on the first four time intervals.

**Figure 3 ijms-21-06602-f003:**
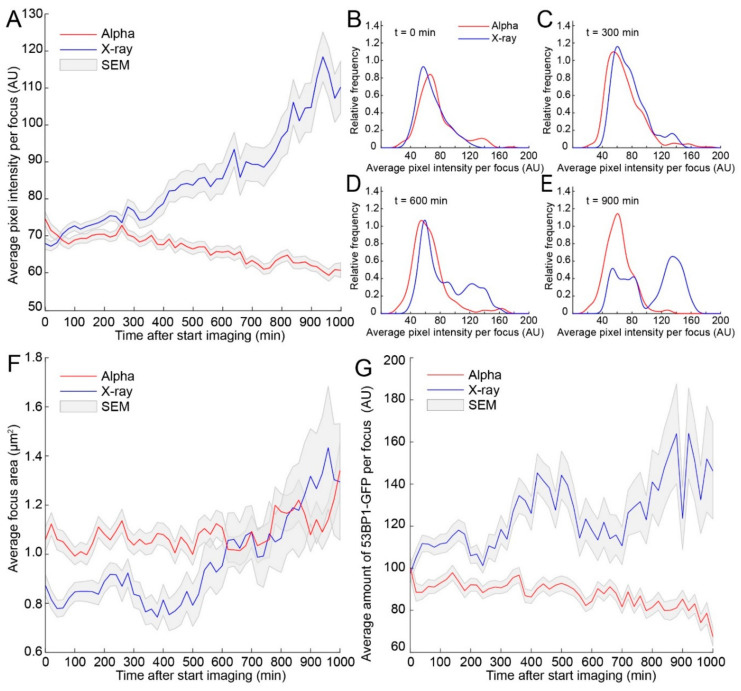
53BP1-GFP protein accumulation and focus size increase after X-ray irradiation. (**A**) Average 53BP1-GFP intensity per focus (average pixel intensity) over time after treatment using α-particle (red) or X-ray (blue) irradiation (2 Gy). (**B**–**E**) Overview of focus population distribution at 0, 300, 600, and 900 min after the start of imaging. Graphs shown kernel density estimations with the area under the curve being equal to the area of the histogram. The 0 AUs are the consequence of the smoothing procedure used for density estimations. (**F**) Average 53BP1-GFP focus area trough time. The average of foci induced by α-particles (red) or foci induced by X-ray (blue). (**G**) Total amount of 53BP1-GFP (product of focus area and pixel intensity) on foci through time after α-particle (red) or X-ray (blue) irradiation. SEM is indicated in gray.

**Figure 4 ijms-21-06602-f004:**
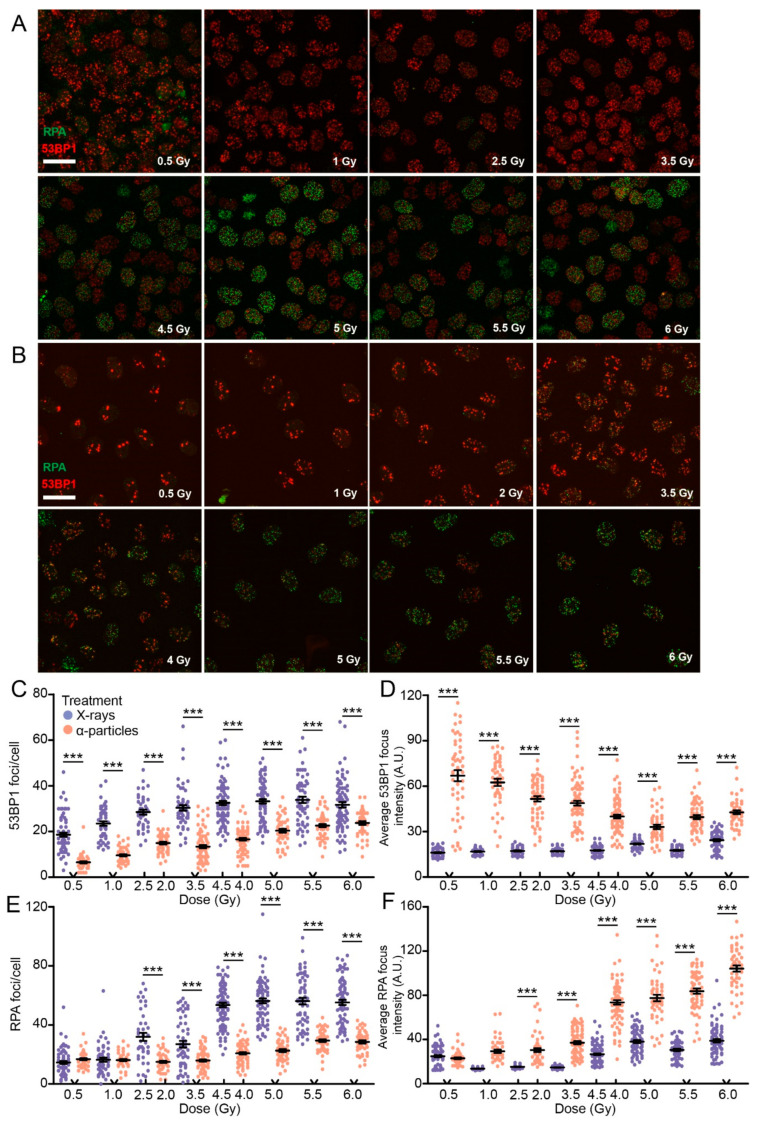
Focus kinetics of 53BP1 and RPA are dose- and radiation type-dependent. Overview of 53BP1 and RPA staining after X-ray (**A**) and α-particle (**B**) irradiation using a dose gradient. Cells were fixed 1 h after irradiation. Quantification of 53BP1 foci per EdU-positive cell (**C**) and intensity (**D**). Quantification of RPA foci per EdU-positive cell (**E**) and intensity (**F**). More than 100 EdU-positive cells were analyzed in two independent experiments. Data points in the plot indicate single nuclei treated with X-ray irradiation (purple) or α-particles (orange). Error bars indicate SEM. Black bars indicate the mean. The statistical differences are indicated by asterisks (***, *p* < 0.001) and determined by ANOVA followed by Tukey’s multiple comparison test.

**Figure 5 ijms-21-06602-f005:**
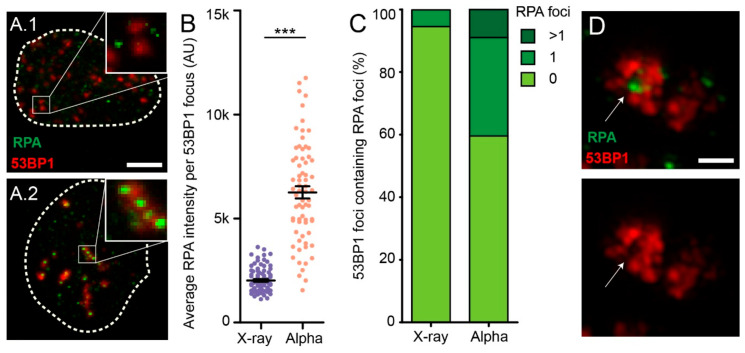
Multiple individual resection events after α-particle irradiation. Representative confocal images of induced 53BP1 and RPA foci by X-ray (**A.1**) or α-particles (**A.2**). Scale bar indicates 5 µm. (**B**) Quantification of RPA focus intensity. (**C**) Percentage of 53BP1 foci colocalizing with 0, 1, or >1 RPA foci. In total, 15 EdU-positive cells were analyzed using structured illumination microscopy for each treatment. (**D**) Representative images of 53BP1 exclusion at RPA foci after α-particle irradiation. Scale bar indicates 0.8 µm. Per assay, >100 EdU positive cells were analyzed. Error bars indicate SEM. The statistical differences are indicated by asterisks (***, *p* < 0.001) and determined by student’s *t*-test.

**Figure 6 ijms-21-06602-f006:**
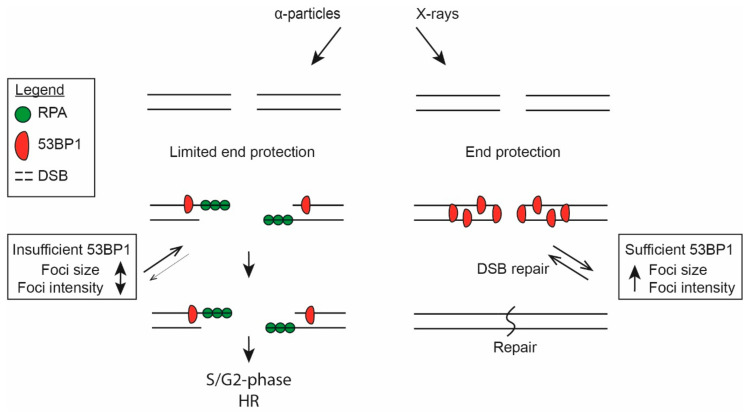
Working model of the role of radiation type in double-strand break (DSB) repair pathway choice. Both X-ray irradiation and α-particle irradiation induce DSBs. The DNA ends induced by X-ray irradiation are protected by 53BP1 and are repaired by the nonhomologous end-joining (NHEJ) machinery. Repair of DSBs reintroduces 53BP1 back into the general pool and causes the remaining foci to recruit more 53BP1, leading to an increase in focus intensity. Repair via the slow component of NHEJ involves chromatin decondensation, leading to focus growth. On the other hand, after α-particle irradiation, the 53BP1 pool is insufficient, thereby limiting end protection and leading to resection. Resection in the S/G2 phase would activate homologous recombination (HR)-directed repair.

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
