# Peer review of "Comparison of High- and Low-LET Radiation-Induced DNA Double-Strand Break Processing in Living Cells"

_ijms, 2020, doi:10.3390/ijms21186602_

Round 1

Reviewer 1 Report

The authors studied dynamics of DNA repair foci in live cells expressing fluorescently tagged 53BP1 after alpha particle and X-ray irradiations. They found certain differences in size and intensity of alpha- and X-ray induced foci, and concluded that alpha-particle induced foci contained multiple DSBs, and that they are more often repaired by homologous recombination pathway.

An article on this subject using the same cells and similar approach was published in this very journal in 2018. Authors referenced this paper (ref. 19). They need clearly state in the Introduction what is the rationale for repeating that previous study. They also should discuss what are their novel findings and why are they different from the previous results.

This reviewer has a major concern regarding the lack of linear increase in the number of foci with the dose for both forms of radiation (Figure 4). The reason for this could be that the authors underestimate the number of foci due the large number of foci per cell. The expected number of foci per cell is 20-40 per Gy. With the resolution they show in Fig. 4, and the lack of 3-D rendering the maximum they can count is ca. 20 foci per cell. They should repeat their studies in the range 0.1 – 1 Gy, where foci could be counted with confidence. Also, at doses higher than 5 Gy there should be significant cell death that could affect DNA damage response.

The radiation doses in all experiments (Figs. 1-3) should be clearly stated in the legends.

The text in the Abstract and Introduction creates impression that live cell foci were also stained for other repair proteins (also line 419 in Methods).

Authors should give rationale for switching from live cells to fixed cells in the experiments in Figures 4 and 5.

The source of 53BP-GFP-U2OS cells should be stated in the Materials and Methods.

The time frame of Supplemental Movie 1 should be presented in the figure legend. If the cell shown in the movie was exposed to 2 Gy one would expect >40 foci. Where are they?

Line 31, Replace “high lethality” to Relative Biological Effectiveness (RBE).

Figure 2 A and B. How to reconcile almost 2 times difference in the number of alpha- and X- foci in panel A and almost no difference in panel B?

This reviewer would recommend staining fixed cells for gamma-H2AX as a gold standard for DNA repair foci.

Author Response

Response to reviewer 1 comments

The authors studied dynamics of DNA repair foci in live cells expressing fluorescently tagged 53BP1 after alpha particle and X-ray irradiations. They found certain differences in size and intensity of alpha- and X-ray induced foci, and concluded that alpha-particle induced foci contained multiple DSBs, and that they are more often repaired by homologous recombination pathway.

An article on this subject using the same cells and similar approach was published in this very journal in 2018. Authors referenced this paper (ref. 19). They need clearly state in the Introduction what is the rationale for repeating that previous study. They also should discuss what are their novel findings and why are they different from the previous results.

We agree with the reviewer and added the rationale and our main findings in the introduction. Lines 75-78.

“A previous study has shown that α-particle induced 53BP1 foci are more persistent, larger and show high mobility compared to X-ray irradiation induced 53BP1 foci. To determine the kinetics and mechanism of DSB processing after high LET a-particle irradiation compared to low LET X-ray irradiation, we repeated the experiment, increased the time frame and investigated DSB repair protein behavior.”

We found that indeed a-particle induced 53BP1 foci were larger and persistent. Interestingly, we found that the remaining foci after X-ray irradiation showed size and intensity increase at the later time-points, which was not observed for a-particle induced foci.

We explain possible reasons why we did not observe differences in mobility between a-particle induced DSBs compared to X-ray induced DSBs in lines 312-322, which was the only difference we found compared to the previous study. In addition, we discuss our new results which we observed with a longer time frame: Size and intensity of X-ray induced foci increase, while these parameters remain constant for a-particle induced foci. Furthermore, both X-ray and a-particle induced foci show decreased mobility with increasing time of presence. In addition with the use of super resolution microscopy we show multiple resection events within one 53BP1 focus.

This reviewer has a major concern regarding the lack of linear increase in the number of foci with the dose for both forms of radiation (Figure 4). The reason for this could be that the authors underestimate the number of foci due the large number of foci per cell. The expected number of foci per cell is 20-40 per Gy. With the resolution they show in Fig. 4, and the lack of 3-D rendering the maximum they can count is ca. 20 foci per cell. They should repeat their studies in the range 0.1 – 1 Gy, where foci could be counted with confidence. Also, at doses higher than 5 Gy there should be significant cell death that could affect DNA damage response.

In our opinion the number of DSBs does not have to be directly proportional to the number of foci present in the nucleus. It is still under debate how many DSBs are represented by one focus. Furthermore, chromatin context, repair kinetics and time of analysis after irradiation all influence the number of foci that are detected. We provide references of recent studies in which DSB clusters within γH2AX or 53BP1 foci were investigated in high resolution, arguing that a 1:1 correlation of the number of DSBs and foci is still under debate [1-3].

The range of 0.1-1 Gy would be ideal for foci counting with confidence using X-ray irradiation. This argument does not hold for a-particle, due to the fact that the chance of an a-particle hitting a nucleus is quite low. For more information on how the a-particle dose is calculated we would like to refer to the following paper [4].

Furthermore, cell death, as a consequence at doses higher than 5 Gy, does not play a role in the time frame we investigated (1 hour post irradiation).

The radiation doses in all experiments (Figs. 1-3) should be clearly stated in the legends.

The used dose is added to all experiments.

The text in the Abstract and Introduction creates impression that live cell foci were also stained for other repair proteins (also line 419 in Methods).

To avoid confusion we added that we use immunostaining for endogenous 53BP1, RPA and RAD51 in sections indicated by Abstract, Introduction and Methods.

Authors should give rationale for switching from live cells to fixed cells in the experiments in Figures 4 and 5.

We adjusted the text in the results accordingly to explain our rationale. Lines 211-212

“In addition, to investigate functional protein and minimize the interference of overexpression constructs cells were stained for endogenous 53BP1 and RPA as markers for DSBs and resection, respectively (Figures 4A and B).”

In order to investigate functional protein in DSB repair pathways we chose for immunofluorescence staining to reduce the possible artefacts connected to over-expression or knock-in strategies, which was previously found for mRad51 [5].

The source of 53BP-GFP-U2OS cells should be stated in the Materials and Methods.

The correct source of the 53BP1-GFP U2OS cell line is referred to with reference 30 indicated in the M&M section 4.1.

The time frame of Supplemental Movie 1 should be presented in the figure legend. If the cell shown in the movie was exposed to 2 Gy one would expect >40 foci. Where are they?

The time frame of the movie has been added to the figure legend. We would like to refer to our answer at point 2 for the additional question.

Line 31, Replace “high lethality” to Relative Biological Effectiveness (RBE).

We agree with the reviewer and ‘high lethality’ was changed to “increased Relative Biological Effectiveness (RBE) compared to X-ray irradiation”.

Figure 2 A and B. How to reconcile almost 2 times difference in the number of alpha- and X- foci in panel A and almost no difference in panel B?

We agree that this figure might cause confusion. In figure 2A, the average number of 53BP1-GFP foci are presented. In figure 2B, we present a percentage of the total amount of tracks with a certain length (duration of track), plotted on the x-axis.

We adjusted the y-axis title with “Percentage of 53BP1-GFP foci tracks”.

To give an example: Almost 70% of the measured 53BP1-GFP foci tracks, induced by alpha particles, have a duration of 20-80 minutes. While more than 75%, induced by X-ray irradiation, have a duration of 20-80 minutes.

This reviewer would recommend staining fixed cells for gamma-H2AX as a gold standard for DNA repair foci.

Both γH2AX and 53BP1 are markers for damaged chromatin where yH2AX is commonly used as marker for initial chromatin modification after DSB induction, 53BP1 marks the subsequent event in chromatin modification. In addition, 53BP1 can be marked for live-cell imaging where yH2AX cannot. Therefore, we chose 53BP1 as DNA damage marker for all experiments.

  1. Bobkova, E.; Depes, D.; Lee, J.H.; Jezkova, L.; Falkova, I.; Pagacova, E.; Kopecna, O.; Zadneprianetc, M.; Bacikova, A.; Kulikova, E., et al. Recruitment of 53bp1 proteins for DNA repair and persistence of repair clusters differ for cell types as detected by single molecule localization microscopy. Int J Mol Sci 2018, 19.
  2. Hausmann, M.; Wagner, E.; Lee, J.H.; Schrock, G.; Schaufler, W.; Krufczik, M.; Papenfuss, F.; Port, M.; Bestvater, F.; Scherthan, H. Super-resolution localization microscopy of radiation-induced histone h2ax-phosphorylation in relation to h3k9-trimethylation in hela cells. Nanoscale 2018, 10, 4320-4331.
  3. Scherthan, H.; Lee, J.H.; Maus, E.; Schumann, S.; Muhtadi, R.; Chojowski, R.; Port, M.; Lassmann, M.; Bestvater, F.; Hausmann, M. Nanostructure of clustered DNA damage in leukocytes after in-solution irradiation with the alpha emitter ra-223. Cancers (Basel) 2019, 11.
  4. Kouwenberg, J.J.M.; Wolterbeek, H.T.; Denkova, A.G.; Bos, A.J.J. Fluorescent nuclear track detectors for alpha radiation microdosimetry. Radiat Oncol 2018, 13, 107.
  5. Uringa, E.J.; Baldeyron, C.; Odijk, H.; Wassenaar, E.; van Cappellen, W.A.; Maas, A.; Hoeijmakers, J.H.; Baarends, W.M.; Kanaar, R.; Essers, J. A mrad51-gfp antimorphic allele affects homologous recombination and DNA damage sensitivity. DNA Repair (Amst) 2015, 25, 27-40.

Reviewer 2 Report

Overall the work presented by Roobol et al., is of a very high quality. The experimental design is sound, and sufficiently well described. The authors have focused on a topic of interest, and have used cutting edge techniques to pursue stategies that permit them to provide some new insight into molecular differences in the cellular/molecular response to high vs low linear energy transfer (LET) radiation.

Of concern is that, with the exception of the data presented in Figure 5, no statistical analysis of apparent differences observed in cells treated with high vs. low LET radiation has been performed. The authors present the results 'at face value', ie. implicitly concluding that apparent differences are significant. The manuscript would be substantially improved were statistical analyses performed on all the data presented. An additional concern involves a number of instances where the authors' interpretations don't appear to strictly agree with the data. For instance in Figure 2A, the authors conclude that 'the number of foci per cell in X-ray irradiated cells showed a decline to 3 foci/nucleus from 2 hours after irradiation onwards. This is at odds with the data presented showing in excess of 5 foci/nucleus at three hours post imaging. (It would also have been extremely helpful to have included data on foci/nucleus prior to irradiation.) In addition, the authors' description of the data presented in figure 4 was very hard to follow, with apparent conclusions being reached based on often very small differences observed at seemingly arbitrary time points comparing the high/low LET-treated cells.

The abstract contains two sentences that are not supported by the data, and in my view should be deleted. The first is  'We conclude that high LET alpha particles cause closely interspaced DSBs leading to high local concentrations of repair proteins'. The second is 'Our results point towards the depletion of soluble protein in the nucleoplasm. . . '. Both are potentially true-and are not contradicted by the data. However, since neither is directly supported by data presented in the text, it seems to me it would be better move them to the discussion-where comments of that nature are appropriate.

Reviewer 3 Report

Roobol et al. investigate the dynamics and formation of 53BP1 foci in the context of DSB end resection and compare these foci to those induced by alpha-particles and X-ray irradiation. This is an interesting study and the results are well consistent with evidence provided in previous studies. I have only a few comments, questions, and suggestions.

  1. Some information on linear energy transfer (LET) is missing. For example, the authors should state what keV/μm was used in the study.

  1. Were alpha-particles irradiated horizontally or vertically? This should also be clarified in the manuscript, although the details might be stated in Roobol et al. 2019 [47].

  1. The title “immunohistochemistry” sounds strange. Immunofluorescence would be more appropriate.

  1. The graph in Figure 2A is rather confusing. The bars for “Alpha” and “X-ray” should be shown separately.

  1. In Figure 4C and E, the y-axis is labeled “Average 53BP1 or RPA foci/cell.” Is this correct? How did the authors generate the scatter plot using the average 53BP1 foci? The labels of the y-axes in other figures should also be carefully checked.

  1. The authors quantified RPA foci in EdU positive cells. However, EdU positive cells are S phase cells, but not G2 cells. Therefore, in the Results section, the authors should carefully explain which cell types were examined.

  1. Similar to Figure 5C, “53BP1 foci containing RAD51 foci” should be examined.

  1. In Figure 6, I suspect that alpha-particles with this LET induce RPA foci in the G1 phase. For example, in Figure 4, there are some cells that do not contain RPA foci. These might be G1 cells. In my understanding, a very high LET is required to induce RPA foci in G1 cells. The response of G1 repair should either be illustrated separately with an additional explanation above or “G1-phase Persistent foci” should be deleted from Figure 6.

  1. I do not think the phrase “dead-end product” is the correct word. This should be rephrased.

Author Response

Roobol et al. investigate the dynamics and formation of 53BP1 foci in the context of DSB end resection and compare these foci to those induced by alpha-particles and X-ray irradiation. This is an interesting study and the results are well consistent with evidence provided in previous studies. I have only a few comments, questions, and suggestions.

Some information on linear energy transfer (LET) is missing. For example, the authors should state what keV/μm was used in the study.

We added the LET of the used a-particles as calculated before [1], which was 115±10 KeV/um. For X-ray we complemented the working voltage of the used X-ray cabinet (Xstrahl Live Sciences) as additional information.

Were alpha-particles irradiated horizontally or vertically? This should also be clarified in the manuscript, although the details might be stated in Roobol et al. 2019 [47].

Indeed, Roobol et al. 2019 states the details about the irradiation procedure. However, we added to the M&M that the cells were irradiated with alpha particles passing vertically through the nucleus.

The title “immunohistochemistry” sounds strange. Immunofluorescence would be more appropriate.

We agree with the reviewer and adjusted this accordingly.

The graph in Figure 2A is rather confusing. The bars for “Alpha” and “X-ray” should be shown separately.

We adjusted figure 2A and now show separated bars for Alpha and X-ray to avoid confusion.

In Figure 4C and E, the y-axis is labeled “Average 53BP1 or RPA foci/cell.” Is this correct? How did the authors generate the scatter plot using the average 53BP1 foci? The labels of the y-axes in other figures should also be carefully checked.

We thank the reviewer for noticing. Indeed, ‘average’ is not the correct word to explain the scatterplot. However, we also plot averages in the same graph. In this way wanted to show both at the same time.

We adjusted the labels of the y-axis in figure 4 to “53BP1 or RPA foci/cell” to avoid further confusion.

The authors quantified RPA foci in EdU positive cells. However, EdU positive cells are S phase cells, but not G2 cells. Therefore, in the Results section, the authors should carefully explain which cell types were examined.

We agree with the reviewer and adjusted all references to “EdU-positive cells”. In section 4.3 we explain that EdU was added for 30 minutes, marking S-phase cells.

Similar to Figure 5C, “53BP1 foci containing RAD51 foci” should be examined.

Due to current technical limitations we were not able to investigate RAD51 in combination with 53BP1 using SIM. To avoid further confusion we chose to remove all RAD51 data from the manuscript.

In Figure 6, I suspect that alpha-particles with this LET induce RPA foci in the G1 phase. For example, in Figure 4, there are some cells that do not contain RPA foci. These might be G1 cells. In my understanding, a very high LET is required to induce RPA foci in G1 cells. The response of G1 repair should either be illustrated separately with an additional explanation above or “G1-phase Persistent foci” should be deleted from Figure 6.

We agree with the reviewer and deleted “G1-phase Persistent foci” from Figure 6.

I do not think the phrase “dead-end product” is the correct word. This should be rephrased.

We rephrased “dead-end product” to ‘Persistent DSBs’.

  1. Kouwenberg, J.J.M.; Wolterbeek, H.T.; Denkova, A.G.; Bos, A.J.J. Fluorescent nuclear track detectors for alpha radiation microdosimetry. Radiat Oncol 2018, 13, 107.

Round 2

Reviewer 1 Report

The authors adequately addressed all my questions.

Reviewer 2 Report

The authors have satisfactorily addressed the concerns raised in the previous review.

Reviewer 3 Report

All of my concerns is adequately amended by the authors.